# Improved MLP Energy Meter Fault Diagnosis Method Based on DBN

Chaochun Zhong [1], Yang Jiang [2], Limin Wang [2], Jiayan Chen [1], Juan Zhou [1], Tao Hong [1] and Fan Zheng [2,*]

1   College of Quality & Safety Engineering, China Jiliang University, Hangzhou 310018, China
2   Zhejiang Institute of Metrology, Key Laboratory of Energy and Environmental Protection Measurement of Zhejiang Province, Hangzhou 310007, China
*   Correspondence: zhengfan6162@163.com

**Abstract:** In order to effectively utilize the large amount of high-dimensionality historical data generated by energy meters during operation, this paper proposes a DBN-MLP fusion neural network method for multi-dimensional analysis and fault-type diagnosis of smart energy meter fault data. In this paper, we first use DBN to strengthen the feature extraction ability of the network and solve the problem of many kinds of feature data and high dimensionality of historical data. After that, the processed feature information is input into the MLP neural network, and the strong processing ability of MLP for nonlinear numbers is used to solve the problem of weak correlation among data in the historical data set and improve the accuracy rate of faults diagnosis. The final results show that the DBN-MLP method used in this paper can effectively reduce the number of training iterations to reduce the training time and improve the accuracy of diagnosis.

**Keywords:** multi-layer perceptions; deep belief network; faults diagnosis; smart energy meters

## 1. Introduction

In the event of a fault, the meter depends largely on the immediate attention of the operation and maintenance personnel. When operations and maintenance personnel arrive, it is often difficult to problem-solve the situation on-site and address the situation that caused the fault—that is, it is easy to repair the faults but cannot know the cause of the faults. The diagnosis of the faults often depends on the subjectivity and one-sidedness of the operation and maintenance personnel. Design solutions, component selection, and process flow vary from manufacturer to manufacturer. At the same time, coupled with the installation location, weather, and other external factors, it may lead to smart energy meters having a complex and diverse set of possible fault causes [1]. On the other hand, with the development of technology, the functions of smart energy meters are becoming richer and richer, which also adds more types of faults and increases the difficulty of fault diagnosis. However, the abundance of functions also brings with it the abundance of data that can be collected by the smart energy meter [2]. In order to effectively reproduce the environment in which the faults occurred and to effectively and accurately diagnose the type of faults in the energy meter, this paper makes full use of the wide variety of data generated during the operation of smart energy meters for the analysis and diagnosis of fault types based on these data.

In previous research on fault diagnosis based on power systems, there are generally non-intrusive remote detection and location for fault diagnosis by using a super-state hidden Markov model (SHM) as mentioned in the literature [3]. However, the method has improved diagnostic accuracy for only a single fault type. In the literature [4], a modified probabilistic Petri net theory is used to diagnose grid faults, which can effectively reduce the unnecessary modeling process and diagnosis time and give correct diagnosis results with incomplete information. However, the effectiveness of this method depends to a large

extent on the appropriateness of the choice of weights, and there are certain difficulties in its practical implementation.

With the development of deep learning technology, its application in the field of fault diagnosis is also increasing. In the literature [5], a deep belief network-based faults diagnosis model for metering devices is proposed to determine the cause of faults for high-magnitude grid anomaly data. The literature [6] fuses all channel features and spatial features to construct a channel-space attention mechanism, forming a feature enhancement module embedded in a sequence model based on selective kernel convolution and deep residual networks combined with multi-layer feature fusion information. Compared with traditional deep learning methods, the model can extract fault features from vibration signals more effectively and improve the recognition rate of faults. The paper shows that reinforcing the features of the dataset before input into the classification neural network is a proven method for fault recognition rate improvement. In the literature [7], an adaptive deep confidence network based on particle swarm optimization is constructed; principal component analysis (PCA) is embedded into the network to reduce the feature dimensionality and is used for bearing fault diagnosis to promote faster convergence. A multi-layer perceptron in a neural network can learn nonlinear relationships through nonlinear hidden layers. This allows it to be applied to many complex datasets and to handle high-dimensional datasets efficiently. The literature [8] used a multi-scale multi-layer perceptron (MSMLO) to achieve the identification of faults in a noisy and small sample train-bearing dataset. The literature [9] used three networks, namely, deep belief network, convolutional neural network, and multi-layer perceptron, to select the network with the best accuracy as a result. However, its essence only exploits the characteristics of a single network and does not integrate the characteristics of multiple networks well. A hybrid CNN-MLP model is proposed in the literature [10] to enhance the extraction of multi-level features and abstract features in a dataset by exploiting the feature that a CNN comprises multiple convolutional neural networks. However, CNN is more suitable for processing image processing and target detection data. In the literature [11], a GRU-MLP faults diagnosis model with an attention mechanism is proposed to enhance the faults diagnosis capability based on time series data and improve the accuracy of MLP in diagnosing faults with correlated characteristics in time. However, DBN still maintains a strong feature extraction capability for high dimensionality, has good compatibility with other algorithms, and its excellent nonlinear mapping capability can better extract the faults information hidden in the original signal [12,13]. The DBN-MLP method was introduced in the literature [14], is applied to trend forecasting, and can be effectively applied to trend forecasting in chaotic time series. The DBN-MLP method is applied in stock forecasting in the literature [15], which shows that DBN can effectively extract hidden features in data, while MLP is effective in processing dense numerical structures and is suitable for handling high dimensional data. However, the DBN-MLP methods mentioned in the above papers are only used in trend forecasting.

The fault data of smart energy meters studied in this paper have 136 relevant characteristic parameters such as manufacturer, device type, asset number, date of commissioning, device status, date of faults discovery, source of the faults, and operating hours. Not only are there multiple types of faults in the fault data, but the faults are also discrete and weakly correlated, and it is troublesome to extract feature information manually. One neural network alone is not a good solution to all the problems that exist in the presence of data. Therefore, this paper proposes a smart energy meter fault diagnosis method based on a DBN-improved multi-layer perceptron network (DBN-MLP) based on the historical fault data of smart energy meters collected from a power system. The problem of high dimensionality and sparse data in fault data is solved by using the powerful adaptive feature extraction capability of DBN. It avoids the problem of too many features and too much computation when using MLP because of the fully connected network, which is too large for high-dimensionality data. It also improves the problem that the fully connected layer of MLP leads to too many weights and complicated calculations under high dimen-

sionality data. After that, the improved data of DBN is input into the MLP network, and the advantages of the MLP network [16] are used to solve the problem of data dispersion and multiple fault type classifications that occur in the historical fault dataset of smart energy meters.

The general structure of this paper is as follows. Section 2 introduces the basic theory of multi-layer perceptron and deep belief networks. In Section 3, the operation flow of the smart energy meter fault diagnosis method based on a multi-layer perceptron with an improved deep belief network and the preprocessing method of the data is described. Section 4 uses the application of the model proposed in this paper to specific data to verify the effectiveness of this paper's method and to compare and analyze the learning results with the other two neural networks to prove the superiority of this paper's method. Section 5 summarizes the research results of this paper and introduces future research directions.

## 2. Basic Theory

### 2.1. Deep Belief Network

A DBN neural network is a deep learning algorithm [17]. It is clear that DBN neural networks can theoretically map arbitrarily complex nonlinear relationships. As a result, the network data do not need to consider the actual physical meaning of each data, and the feature values only consist of numbers. Therefore, a DBN neural network can be selected to classify the historical fault data of smart energy meters. A deep belief network consists of several layers of neurons with a core stack of restricted Boltzmann machine (RBM) layers [18]. The structure is shown in Figure 1.

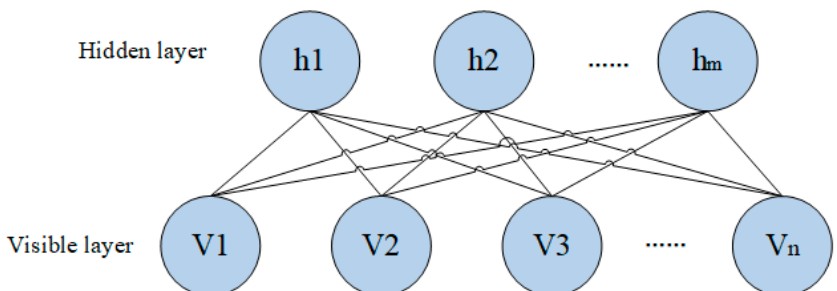

**Figure 1.** Structure diagram.

Its energy function can be expressed as

$$E(v, h|\theta) = -\sum_{i=1}^{n} a_i v_i - \sum_{j=1}^{m} b_j h_j - \sum_{i=1}^{n} \sum_{j=1}^{m} v_i w_{ij} h_j \tag{1}$$

In Equation (1): $\theta = (w_{ij}, a_i, b_i)$ is the RBM parameter, n, m is the number of neurons in the visible and hidden layers, respectively, v is the visible layer input unit vector, $v_i$ is the state of the visible layer neuron $i$, and set its bias value to $a_i$; $h$ is the hidden layer output unit vector, $h_j$ is the state of the hidden layer neuron j, and set its bias value to $b_j$; neuron i and connection weight j weight are defined as $w_{ij}$.

The updated formula for the joint probability distribution of $v_i$ and $h_j$ is

$$p(v, h|\theta) = \exp(-E(v, h|\theta)) / \sum_{v,h} \exp(-E(v, h|\theta)) \tag{2}$$

The DBN feature extraction process is a layer-by-layer learning process with multiple RBM overlays, including forward learning and reverse reconstruction, which can map complex signals to the output and strengthen the features of the output data.

### 2.2. Multi-Layer Perceptron

An MLP neural network is a kind of feedforward neural FFNN [19]. An MLP is characterized by only one implicit layer, one-way connections between neurons, and that data transfers in the MLP occur between three parallel levels: The number of nodes in the input, hidden and output layers are equal to the number of features of the input data, and the number of nodes in the output layer is equal to the number of categories of the output data. Each node in the input layer is connected to all nodes in the hidden layer, and each node in the hidden layer is connected to all nodes in the output layer. The structure is shown in Figure 2 below.

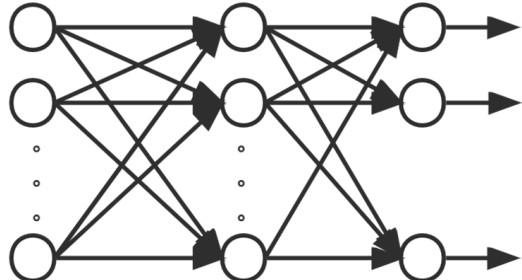

**Figure 2.** MLP structure diagram.

A connection between two nodes has a weight value that indicates the relationship between these two nodes. The hierarchical connection has a weight property, and the node function can perform both summation and activation functions. The summation function is

$$S_j = \sum_{i=1}^{n} w_{i,j} I_i + \beta_j \tag{3}$$

where $n$ is the amount of input data, $I_i$ is the input data, $\beta_j$ is the deviation, and $w_{i,j}$ is the connection weight.

The output is obtained in the hidden layer using the activation function as

$$f_j(x) = \frac{1}{1 + e^{-s_j}} \tag{4}$$

The output of the output layer cell in the MLP can be obtained by combining Equations (3) and (4)

$$y_i = f_j \left( \sum_{i=1}^{n} w_{i,j} I_i + \beta_j \right) \tag{5}$$

### 3. DBN-MLP Fault Diagnosis Method

The fault diagnosis process of the smart energy meter based on the DBN-MLP model is shown in Figure 3:

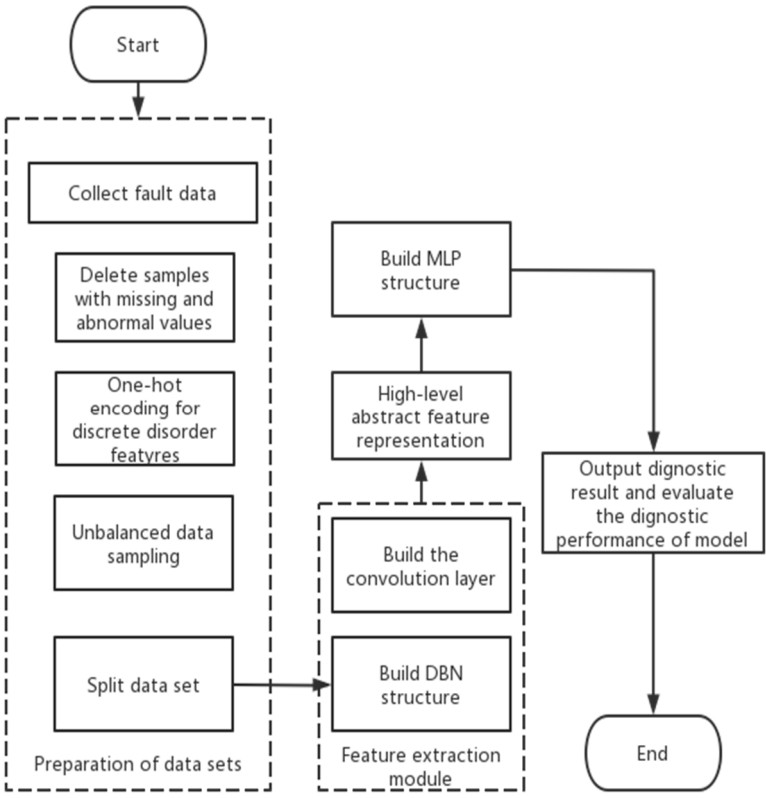

**Figure 3.** Smart energy meter fault diagnosis process based on the DBN-MLP model.

Step 1: Collect the historical fault data of smart energy meters and form the data set.

Step 2: Preprocessing of faulty sample data. Remove the faulty samples with missing values and abnormal values; one-hot encoding of discrete and disordered feature attribute values [20].

Step 3: Perform imbalanced data sampling and partitioning of the data set. Since the historical data set of smart energy meters is unbalanced-like data, a hybrid sampling method combining oversampling and undersampling is required. Random sampling is performed for fault types that require undersampling, and SMOTE sampling is performed for fault types that require oversampling, and the formula for determining the theoretical sample size after sampling for each fault type is shown in Equation (6).

$$N\_new_i = \begin{cases} N_i - a * (N_i - N_{mean})(undersampling) \\ N_i + a * (N_{mean} - N_i)(oversampling) \end{cases}, i = 1, 2, \ldots K \tag{6}$$

Suppose there are K types of faults in the pre-processed smart energy meter fault data set; $N\_new_i$ denotes the number of samples after sampling for category i, $N_i$ denotes the number of samples before sampling for category i, and a denotes the sampling balance coefficient. Here, we take a = 0.5, $N_{med}$ means the median number of samples of all categories before sampling $N_i$; if greater than $N_{med}$, takes the undersampling method and vice versa takes the oversampling method.

The data set is divided into a training set and a test set for training the parameters of the DBN-MLP model and the performance of the faults diagnosis model.

Step 4: Construct the DBN structure and set network parameters to achieve faults feature extraction. The trained feature representation is passed from the upper-layer RBM to the lower-layer RBM, and finally, the high-level abstract feature representation of the input data is obtained.

Step 5: The DBN-MLP model is trained using a multi-layer neural network with a nonlinear activation function in MLP to train the mapping between high-level abstract feature representations and categories. In MLP, there is a set of weights and bias terms

between each hidden layer, which perform linear transformations and nonlinear activation function transformations on the input data. Finally, the MLP will use an output layer for classification prediction. Calculate the loss function value (error) from the diagnostic results and the actual results. In the reverse weight update process, the error is first passed from the output layer to the intermediate layers using the chain rule, and then the weights of each layer are updated by the gradient descent method. Training will be stopped when the loss value changes below 0.001.

Step 6: Output the diagnostic results and evaluate the diagnostic performance of the DBN-MLP model using the test set. The performance evaluation metrics are accuracy (A), precision (P), recall (R), and F-value. Suppose there are k fault types and $n_{ij}$ denotes the number of samples that diagnose type i to type j. Then there are

$$A = \sum_{i=1}^{k} n_{ii} / \sum_{i=1}^{k} \sum_{j=1}^{k} n_{ij} \tag{7}$$

$$P_i = n_{ii} / \sum_{j=1}^{k} n_{ji} \tag{8}$$

$$R_i = n_{ii} / \sum_{j=1}^{k} n_{ij} \tag{9}$$

$$F_i = 2P_i R_i / (P_i + R_i) \tag{10}$$

$$Macro\ F1 = \frac{1}{k} \sum_{i=1}^{k} F_i \tag{11}$$

From Equation (12), accuracy (A) is the ratio of the number of samples correctly predicted by the classifier to the total number of samples, and the higher the accuracy, the greater the ratio of the number of samples correctly predicted by the classifier to the total number of samples. For multi-classification problems, accuracy itself may be affected by the imbalance of the data set in practical applications. Therefore, the performance of the classifier is usually also evaluated by considering the precision (P), recall (R), and F-value metrics. Precision refers to how many of the samples predicted to be positive are truly positive, and the higher the precision, the greater the proportion of samples predicted to be positive by the classifier; recall is a measure of the coverage of a category of diagnostic results by a classification model; F-value is a comprehensive metric for classification models that includes precision and recall, and can effectively evaluate models that have requirements for both precision and recall.

## 4. Example Verification

### 4.1. Fault Dataset Preparation of Smart Energy Meters

The data in this article were provided by Dr. Zhou, and we sincerely thank her for providing the data [21]. The fault data of smart energy meters studied in this paper have 136 relevant characteristic parameters such as manufacturer, equipment type, asset number, date of commissioning, equipment status, faults discovery date, faults source, operating hours, operating time, power supply unit, equipment specification, and communication method. In this paper, the fault diagnosis of smart energy meters is intended to restore as much as possible all the potential factors affecting the electrical energy faults in order to be able to increase the accuracy of the faults diagnosis. The faults of the meter may be due to other non-environmental factors, such as the faults of a batch of the manufacturer's products, in addition to the prevailing weather conditions or the influence of electricity. By combining the data sets under various factors, the accuracy and confidence of the fault diagnosis can be improved more effectively. Six types of faults corresponding to the fault data were obtained from the enterprise data, namely: overload burn meter, battery faults, pulse sampling faults, clock faults, communication faults, and electromechanical faults.

The fault types are numbered 0–6 in ascending order of the number of samples. In the sample data, there are discrete and disordered feature variables such as equipment type, equipment status, manufacturer, etc. This paper uses one-hot coding to digitize the discrete and disordered features for better deep learning.

The histogram of the distribution of the number of faulty samples is plotted, as shown in Figure 4.

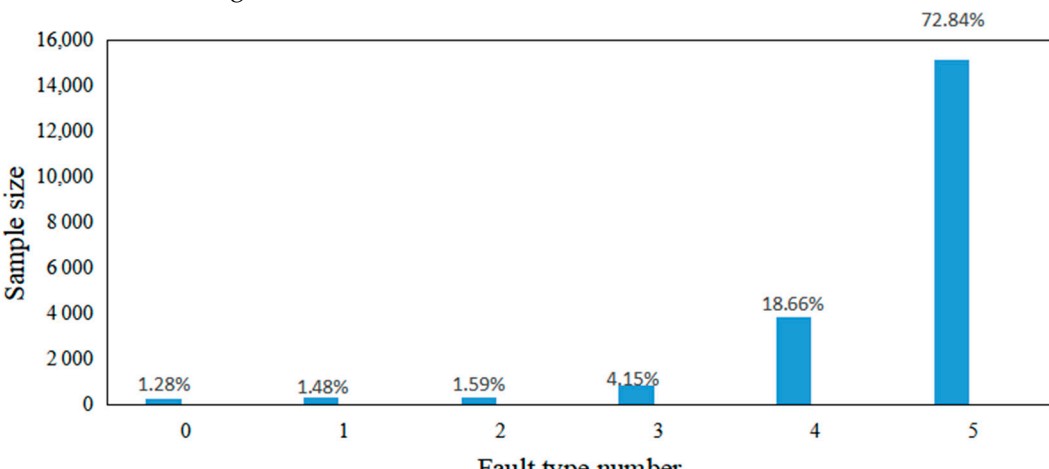

**Figure 4.** Histogram of the distribution of sample sizes for different fault types.

As can be seen in Figure 4 above, the sample size of each fault type is unbalanced, with fault types 4 and 5 accounting for a larger proportion—nearly 92% combined—but fault types 0 to 3 accounting for only about 8%. According to Pareto's law, when a classification accounts for more than 80% of the total classification, it is obvious that it will easily be considered the most dominant fault type, making the classification model ignore small sample types and greatly affecting the fault diagnosis results.

Therefore, this paper uses a hybrid sampling method of oversampling and undersampling, i.e., random sampling for fault types that need to be undersampled and the SMOTE sampling method for fault types that need to be oversampled.

$$\text{New Sample } X' = X + \alpha(X_n - X) \tag{12}$$

For a minority class sample X, find its k nearest neighbor samples $X_1, X_2 \ldots, X_k$. The larger the value of k, the less the sample is guaranteed to be true, so generally, set the value of k to X. For each minority class sample, Euclidean distance from the shortest distance randomly selects the nearest neighbor sample $X_n$, and then two samples X and $X_n$, using a uniform distribution randomly select a weight $\alpha \in [0,1]$, it is possible to increase the number of minority class samples by computing a new sample X'.

The sample size after mixed sampling of unbalanced data is shown in Table 1 and Figure 5.

**Table 1.** Change in sample proportion after mixed sampling method.

| Fault Type Number | Fault Type | Sample Size before Sampling | Sample Size after Sampling | Proportion before Sampling | Proportion after Sampling |
|---|---|---|---|---|---|
| 0 | Overload burn-out meter | 267 | 1865 | 1.28% | 8.97% |
| 1 | Battery faults | 307 | 1885 | 1.48% | 9.07% |
| 2 | Pulse sampling faults | 330 | 1897 | 1.59% | 9.13% |
| 3 | Clock out of order | 863 | 2163 | 4.15% | 10.41% |
| 4 | Communication faults | 3879 | 3671 | 18.66% | 17.66% |
| 5 | Electromechanical faults | 15,142 | 9303 | 72.84% | 44.76% |

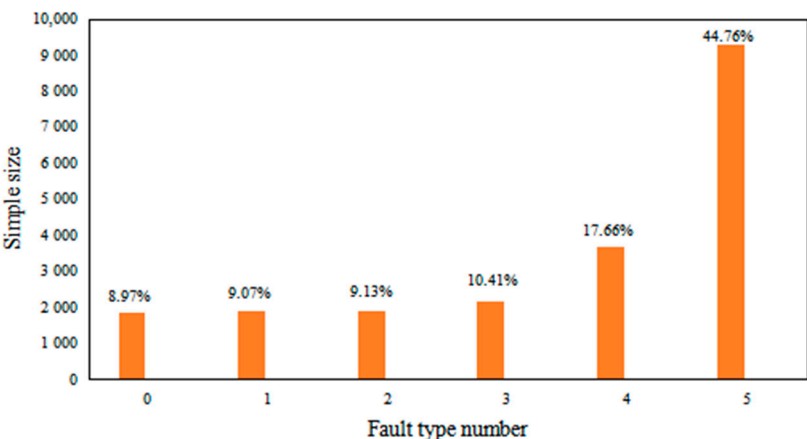

**Figure 5.** Distribution of fault samples after mixed sampling.

The results of the sample data after the hybrid sampling method processing are shown in Table 1 above. The sample data after processing are more balanced in terms of the number of samples on the six faults than before processing. The direct size reduction of the sample size can effectively increase the training efficiency of the subsequent neural network model for training to avoid overfitting.

*4.2. Performance Verification of DBN-MLP Faults Diagnosis*

In the training process, 80% of the smart energy meters fault dataset is used as the training dataset and 20% is used as the test dataset. After data preprocessing, the data are input into the DBN, and the number of DBN neurons is equal to the number of feature dimensions of the dataset, and the number of hidden layers is 1.5 times the number of input layers in order to learn more features. Finally, a multi-layer perceptron neural network is built, and the number of hidden layers corresponds to the number of output layers of the DBN neural network, and the maximum number of iterations is chosen as 500.

In this paper, the robustness of the model is verified by the cross-validation method, which estimates the robustness of the model by performing multiple random splits on the dataset before training the model. The dataset is partitioned five times, and each partition generates a training set for training and a test set for testing. After each segmentation, the model is trained on the training set and evaluated on the test set. Since the method in this paper is based on a fully connected layer, each of its nodes is connected to the nodes in the next layer. Therefore, in case of noise, the fully connected layer feels more sensitive than the convolutional layer. Figure 6 shows the average performance of each model on different data sets, indicating that the noise is effectively removed in the pre-processing of this paper. The new data adaptation ability is also strong.

From Figure 7 below, it can be seen that both DBN-MLP and DBN-CapsNet enter convergence quickly, and their post-convergence loss values are lower than those of CNN and MLP. However, the post-convergence loss of DBN-CapsNet is 0.113 is 1.67 times that of DBN-MLP. This shows that the faults diagnosis result of DBN-MLP is better than DBN-CapsNet because its predicted result is closer to the real value.

It is shown that the MLP model with DBN enhanced feature extraction capability can better fit the training data during the training process, and its training performance is better than the other three models. Next, the generalization ability of the model is evaluated by a test set. Figure 7 below shows the confusion matrix of the test set faults diagnosis results.

According to Figure 8, three model precision (P), recall (R), and F-values can be further calculated to evaluate the goodness of the model training results, and their specific data are shown in Table 2.

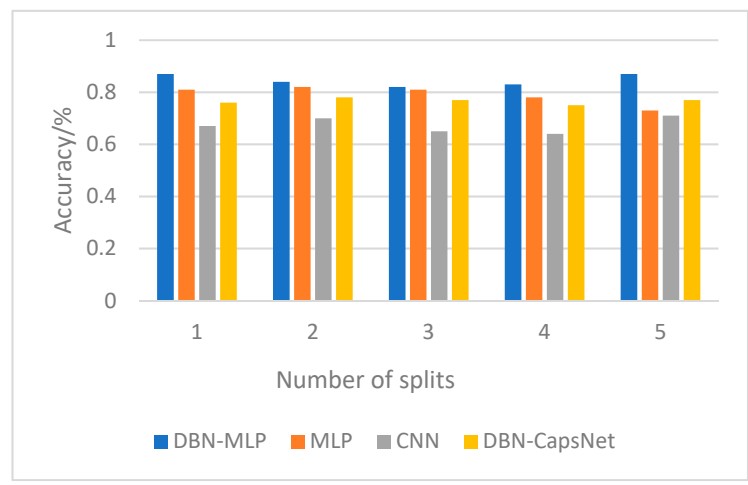

**Figure 6.** Cross-validation results.

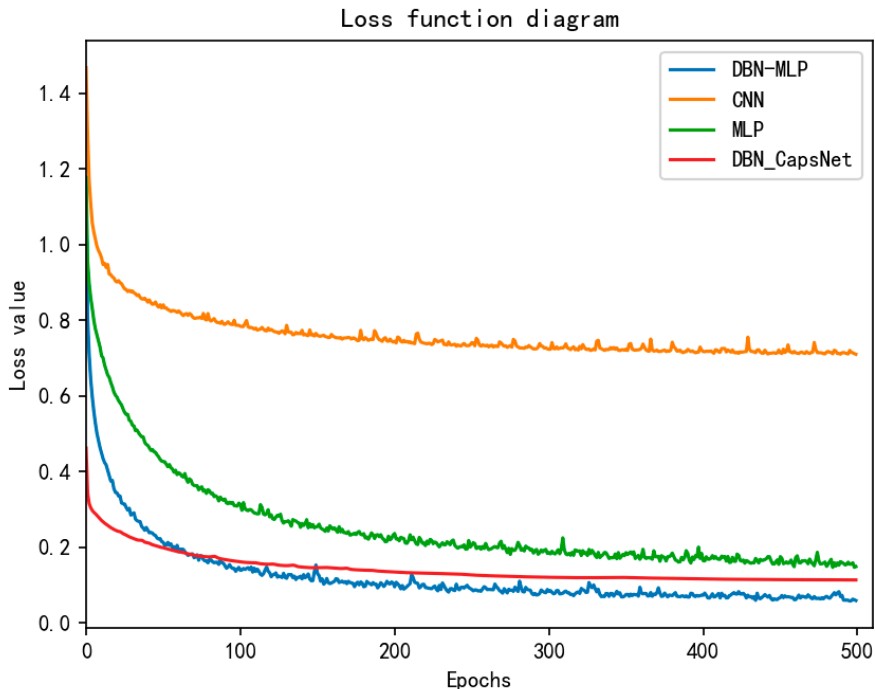

**Figure 7.** Comparison of loss rate convergence.

**Table 2.** The precision rate, recall rate, and F value of MLP, CNN, and DBN-MLP.

| Fault Type Number | MLP | | | CNN | | | DBN-MLP | | | DBN-CapsNet | | |
|---|---|---|---|---|---|---|---|---|---|---|---|---|
| | Precision | Recall | F Value | Precision | Recall | F Value | Precision | Recall | F Value | Precision | Recall | F Value |
| 0 | 0.86 | 0.84 | 0.84 | 0.58 | 0.80 | 0.67 | 0.9 | 0.89 | 0.9 | 0.80 | 0.85 | 0.83 |
| 1 | 0.85 | 0.87 | 0.87 | 0.78 | 0.63 | 0.70 | 0.91 | 0.92 | 0.91 | 0.88 | 0.85 | 0.86 |
| 2 | 0.83 | 0.83 | 0.83 | 0.42 | 0.75 | 0.53 | 0.9 | 0.89 | 0.9 | 0.83 | 0.82 | 0.82 |
| 3 | 0.76 | 0.79 | 0.79 | 0.63 | 0.69 | 0.66 | 0.83 | 0.87 | 0.85 | 0.79 | 0.80 | 0.79 |
| 4 | 0.71 | 0.68 | 0.68 | 0.45 | 0.52 | 0.48 | 0.74 | 0.75 | 0.74 | 0.67 | 0.66 | 0.66 |
| 5 | 0.83 | 0.84 | 0.84 | 0.83 | 0.71 | 0.76 | 0.88 | 0.86 | 0.87 | 0.83 | 0.83 | 0.83 |

From Table 2, it can be seen that the MLP neural network, with further feature enhancement by DBN, has further improved the classification effect for each fault. Unlike CNNs, CNNs obtained precision similar to that of MLP neural networks only in type 5, which has the largest amount of data, because the dataset studied in this paper is an unbalanced dataset. From this, it can be seen that MLP neural network is more suitable than CNN for the classification task in this paper. Precision's better performance indicates that the

paper's treatment of the data sample is reasonable, and the proportion of both positive and negative samples of the data sample is in balance.

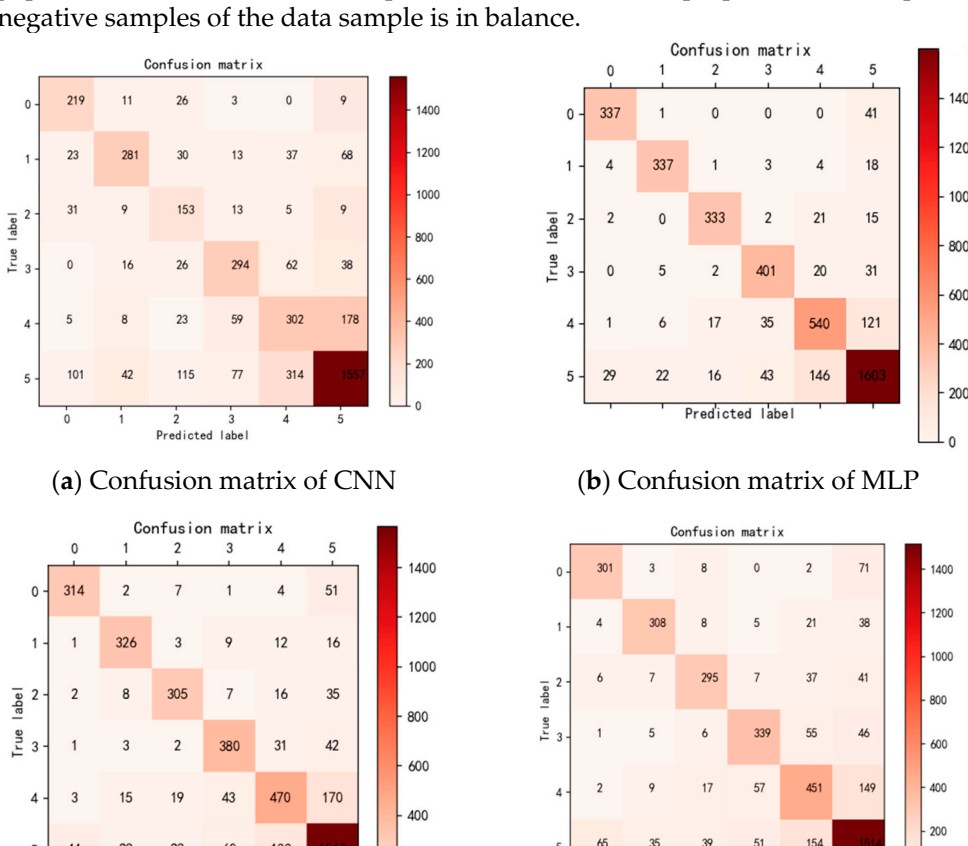

(**a**) Confusion matrix of CNN   (**b**) Confusion matrix of MLP

(**c**) Confusion matrix of DBN-MLP   (**d**) Confusion matrix of DBN-CapsNet

**Figure 8.** Comparison of four model confusion matrices.

While the performance of DBN-CapsNet is similar to that of MLP, overall, it is inferior to that of DBN-MLP. Next, we process the data to see the results of the model's combined performance. The data in Table 2 above were averaged for each classification of faults diagnosis, and the corresponding accuracy and training time were calculated to obtain Table 3 to further evaluate the excellence of the model.

**Table 3.** Comparison of evaluation parameters of three models.

| Model | DBN-MLP | MLP | CNN | DBN-CapsNet |
|---|---|---|---|---|
| Accuracy | 0.85 | 0.81 | 0.68 | 0.77 |
| Mean F value | 0.86 | 0.81 | 0.63 | 0.80 |
| Train time | 150 s | 160 s | 669 s | 2400 s |

The data studied in this paper are historical fault data of smart energy meters, which have high requirements on both precision and recall of the model at the same time, so the classification effect of the model can be evaluated directly by the mean F value. From Table 3 above, we can see that the mean F value of DBN-MLP is 4%, 17%, and 6% higher than that of MLP, CNN, and DBN-CapsNet, respectively. The accuracy of DBN-MLP is 5%, 15%, and 8% higher than that of MLP and CNN and DBN-CapsNet, respectively. Train time is 10 s and 519 s faster than MLP and CNN. It can be seen that the fully connected layer is better than the convolutional layer in terms of both efficiency and accuracy in processing high-dimensional numerical data. Again, the fully connected layer can be trained faster

than the capsule network structures. The comparison of the evaluation parameters shows that the diagnostic results of DBN-CapsNet and MLP are similar, but the training time is much longer than that of MLP. On the other hand, the training time of DBN-MLP proposed in this paper is about the same as that of MLP, but its accuracy and mean F-value are improved compared with both the MLP and the DBN-CapsNet. It also takes into account the training efficiency and the accuracy of the diagnostic results.

The model in this paper is trained using CPU, the specific hardware used is 11th Gen Intel(R) Core(TM) i7-11800H@2.3GHz with 16 GB of RAM and a speed of 3200 MHz. The model occupies the CPU and RAM during training is shown in Table 4 below.

**Table 4.** Computational cost comparison.

| Model | DBN-MLP | MLP | CNN | DBN-CapsNet |
|-------|---------|-----|-----|-------------|
| CPU Occupancy | 65% | 62% | 100% | 100% |
| RAM | 11.4 GB | 11.2 | 11.7 GB | 11.3 GB |

The CPU usage of MLP is smaller than that of CNN and capsule networks because MLP does not have convolutional and pooling layers; it only has fully connected layers and thus needs to compute fewer parameters. The computation of MLP during training and prediction depends mainly on its input data size and the number of hidden layers rather than on its model structure itself. In contrast, CNN and capsule networks both contain convolutional and pooling layers and need to compute more parameters, so they are more computationally intensive than MLP.

## 5. Conclusions

In this paper, a smart energy meter fault diagnosis model based on the DBN-improved multi-layer perceptron (DBN-MLP) is proposed. The main work and results are as follows.

(1) This paper solves the problem of uneven distribution of faults sample data in the dataset and improves the accuracy of faults classification by using a hybrid sampling method combining under-sampling and over-sampling.

(2) Compared with MLP, this paper improves the faults diagnosis capability by using MLP for faults classification after enhancing the features of input data by DBN. At the same time, DBN also solves the problem of too many features and sparse data brought by the high dimensionality, realizes the automatic acquisition of features, and improves the problem that MLP is prone to overfitting under the construction of its fully connected layer. Accuracy and macro F1 improved by 5% and 15%, respectively. The training time and computational cost of DBN-MLP are even less than that of DBN-CapsNet. Overall, DBN-MLP is superior to MLP, CNN, and DBN-CapsNet in terms of training performance and generalization ability for the goal of classifying smart energy meter fault data.

In summary, the DBN-MLP method proposed in this paper provides better improvements in training speed, computational cost, and accuracy than previous faults diagnosis research methods based on historical fault data of smart energy meters. The next step can be to improve the generalizability of the method in this paper and extend the method to more fields of smart energy meters.

**Author Contributions:** Writing-original draft, C.Z.; Data curation, J.Z.; formal analysis, J.C.; investigation, Y.J.; methodology, T.H.; software, F.Z.; Writing—review and editing: L.W. All authors have read and agreed to the published version of the manuscript.

**Funding:** This research was funded by Zhejiang Provincial Market Supervision Administration Scientific Research Project. Research on key technology of full-coverage intelligent test for energy meter EMS test (ZC2021A010).

**Institutional Review Board Statement:** Not applicable.

**Informed Consent Statement:** Not applicable.

**Data Availability Statement:** Not applicable.

**Conflicts of Interest:** The authors declare no conflict of interest.

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
