# Peer review of "Improved MLP Energy Meter Fault Diagnosis Method Based on DBN"

_electronics, doi:10.3390/electronics12040932_

Round 1
Reviewer 1 Report
Please note the following observations after comprehensive review of the paper:
1. Referring to the previously published research work as mentioned below by the authors Zhou J, Wu Z, Wang Q, Yu Z. on “Fault Diagnosis Method of Smart Meters Based on DBN-CapsNet” in Electronics Journal of MDPI:
Zhou J, Wu Z, Wang Q, Yu Z. Fault Diagnosis Method of Smart Meters Based on DBN-CapsNet. Electronics. 2022 May 18;11(10):1603.
it is observed that this manuscript also uses the same dataset for Fault Diagnosis of Smart Meters and one of the authors is common in both the papers. Surprisingly, this paper has not been cited by the authors in this manuscript.
22. In my opinion, authors should compare their proposed DBN-MLP fusion neural network method with their previously published DBN-CapsNet Fault Diagnosis Method using the same metrics Precision(P), Recall(R) and F-value and logically conclude their findings to add to conceptual advancement of the knowledge in the research domain of Fault Diagnosis Method of Smart Meters.
33.The effects of batch size, optimizer, and iteration times on the fault diagnostic performance of smart meters should also be explored as done in the previous work.
44. The performance of proposed methodology should also be compared with the previously proven methodologies for Smart Meter Fault Diagnosis in terms of accuracy as well as computational cost.
55. Manuscript has some grammatical and formatting errors e.g., Ref 8,9,10,11,12,13 14,15,16,17,18 should be written in square brackets. These references are just not visible in the manuscript.
Author Response
Thank you for your suggestion. Modification details are shown in the attachment

Reviewer 2 Report
The main idea of the work is to apply artificial intelligence based on a DBN-MLP fusion neural network method for multi-dimensional analysis and fault type diagnosis of smart energy meter fault data. The work is based on a hybrid architecture for its development, it is also used to solve the problem of weak correlation between the data in the historical data set and improve the accuracy rate of fault diagnosis. The work is interesting as a new form of diagnosis of smart energy meter fault faults
A total of 18 references from different topics related to the research were used in the work, which provides an idea of the updating of the subject matter addressed in the research. However, the introduction needs to be improved and incorporate references of practical examples of DBN-MLP. Since it is a neural structure that has been used in other applications.
The wording of the text needs to be improved. There are concordance problems in phrases and sentences. Revise in its entirety
Example: “in order to be able to increase the Accuracy of the fault diagnosis. And the failure of the meter may be due to other non- environmental factors”
Queries and comments.
1. What is the system response to untested data of the same type of faults?
2. The authors must incorporate images of the experiments, types of sensors, etc. since it is only described very briefly and it is not essentially clear how the data was acquired.
3. What is the computational cost of the proposed method to untested data? make a comparison with some others methods to the same task.
4. What is the purpose of Figure 4, besides to differentiate in percentage value, the failure types in the data?
5. The figure shows the comparison of the proposed algorithm with two other methods. What would happen if the number of outlier and the number of untested data were increased? Can the authors describe the process graphically?
Author Response

(The authors gave the same response as above.)

Round 2
Reviewer 2 Report
The work has improved with the corrections and can be published.
Suggestions:
In Figure 4, if possible, scale up the Y-axis, so that the percentage value can be inside the graph.